# Marker-Independent Food Identification Enabled by Combing Machine Learning Algorithms with Comprehensive GC × GC/TOF-MS

**DOI:** 10.3390/molecules27196237

**Published:** 2022-09-22

**Authors:** Bei Li, Miao Liu, Feng Lin, Cui Tai, Yanfei Xiong, Ling Ao, Yumin Liu, Zhixin Lin, Fei Tao, Ping Xu

**Affiliations:** 1State Key Laboratory of Microbial Metabolism, Joint International Research Laboratory of Metabolic & Developmental Sciences, School of Life Sciences & Biotechnology, Shanghai Jiao Tong University, Shanghai 200240, China; 2Key Laboratory of Systems Biomedicine (Ministry of Education), Shanghai Center for Systems Biomedicine, Shanghai Jiao Tong University, Shanghai 200240, China; 3National Engineering Research Center of Solid-State Brewing, Luzhou 646000, China; 4The Instrumental Analysis Center, Shanghai Jiao Tong University, Shanghai 200240, China

**Keywords:** Chinese liquors, food inspection, GC × GC/TOF-MS, machine learning

## Abstract

Reliable methods are always greatly desired for the practice of food inspection. Currently, most food inspection techniques are mainly dependent on the identification of special components, which neglect the combination effects of different components and often lead to biased results. By using Chinese liquors as an example, we developed a new food identification method based on the combination of machine learning with GC × GC/TOF-MS. The sample preparation methods SPME and LLE were compared and optimized for producing repeatable and high-quality data. Then, two machine learning algorithms were tried, and the support vector machine (SVM) algorithm was finally chosen for its better performance. It is shown that the method performs well in identifying both the geographical origins and flavor types of Chinese liquors, with high accuracies of 91.86% and 97.67%, respectively. It is also reasonable to propose that combining machine learning with advanced chromatography could be used for other foods with complex components.

## 1. Introduction

Quality and safety are among the greatest concerns of food product consumers. Traditional food inspection strategies mainly rely on the determination of one or several marker compounds of a food product, which are non-comprehensive, because there are usually thousands of chemicals contained in even a simple food product [1,2]. Fingerprinting-based studies aim for the comprehensive analysis of numerous non-targeted compounds in a complex matrix and give a more holistic representation of the sample composition, without the specific identification of target analytes [2]. Fingerprinting analysis is regarded as a promising method for analyzing foodstuffs and medicines with complicated compositions, and has been widely used in the quality control of tea [3], wine [4,5], and traditional Chinese medicine [6]. Currently, fingerprinting analysis is mainly based on gas chromatography-mass spectrometry (GC-MS), gas chromatography-olfactometry (GC-O), or the mass spectrometry-based electronic nose (MS e-nose), which can only detect limited numbers of chemicals. Therefore, the development of a fingerprinting analysis method based on comprehensive analysis technology is of much interest.

For the chemical analysis of complex mixtures, the comprehensive two-dimensional gas chromatography coupled with time of flight mass spectrometry (GC × GC/TOF-MS) is one preferred approach, which could provide the best separation ability and high peak capacity [7,8]. It also has the greatest ability to separate trace analytes from the mixtures due to the modulation process and distribution of analytes over a retention plane created by the two independent columns. The unrelated separation mechanisms provided by the two columns result in a much higher peak capacity. The GC × GC/TOF-MS is broadly applied for the identification and quantitation of trace components in complex foodstuffs. In particular, it is an important detection technology in the food industry where volatile analytes responsible for odor may be present at low concentrations [9]. GC × GC/TOF-MS has been applied successfully in the chemical detection of foods, such as tobacco [10], wine [11,12], coffee [13], fruits [14], and tea [15]. Zhu et al. [16] first systematically analyzed and identified 478 volatile compounds as common constituents of 55 Xihu Longjing tea samples, using GC × GC/TOF-MS. In 2015, Yao et al. [16] were able to separate more than 1800 volatile compounds in Maotai liquor using GC × GC/TOF-MS, which demonstrated the value of the GC × GC/TOF-MS analysis in decomposing the complex compound mixtures of Chinese liquor (Baijiu).

For the processing of large chromatographic and/or spectral datasets typically generated within complex matrix analyses, an effective algorithm capable of rapid data mining is critical. Previous studies [10,17] have demonstrated the successful application of multivariate statistical analyses (such as HCA, PCA, and PLS-DA) for extracting useful information from chromatographic and spectral datasets and enabling the construction of classification and prediction models. However, its use becomes problematic when a large number of variables are measured, as the datasets not only become larger in size but have more complex relationships [18]. Advanced machine learning techniques have been well-established and used in information technologies. For example, the artificial neural network (ANN), the support vector machine (SVM), and the random forest (RF) methods have been applied in pattern recognition [18,19]. Of these algorithms, SVMs have been proposed as an excellent method for solving complex nonlinear classification and regression problems.

Chinese liquor is an extensively popular alcoholic beverage with a consumption of more than 4 million kiloliters per year in China [20]. It is a crucial foodstuff and is becoming increasing significant in the national economy. This leads to an increase in the value of Chinese liquor as well as an extensive existence of Chinese liquor adulteration [21,22]. Thus, the detection of the Chinese liquor authenticity of Chinese liquors is very important. It is a complex mixture with a wide range of minor organic and inorganic constituents of different physicochemical properties due to its multitudinous raw materials and different manufacturing processes [20]. Therefore, the quality inspection of Chinese liquors is extremely difficult, especially for the identification of adulteration. It is urgently needed to develop a reliable and convenient method to characterize the chemical composition and the identification of different kinds of Chinese liquors. Moreover, the establishment of a detection method for Chinese liquor may contribute to and provide a guide for the food inspection system.

Many studies have been previously carried out, and various techniques have been applied to the investigation of chemicals in liquors. Traditional gas chromatography-olfactometry (GC-O) and the mass spectrometry-based electronic nose (MS e-nose) could be used to obtain the characteristic volatile patterns of the tested Chinese liquors [21,22]. One-dimensional gas chromatography coupled with mass spectrometry (1D-GC-MS) is a frequently employed tool for the profiling of volatile compounds [4,23]. However, most of these studies were focused on the identification and quantitation of volatile chemicals and were dependent upon marker compounds. This neglected the combination effects of different components and often led to biased results in the practice of food inspection. Moreover, these marker compound dependent methods are easier to be cracked by lawbreakers due to the marker compounds that can be supplemented in liquors. Therefore, a marker-independent method is urgently needed for comprehensively controlling the quality of Chinese liquors.

In this study, we established a marker-independent food identification method. The method integrated advanced chromatographic techniques with machine learning. The developed method was used for the identification of Chinese liquors from 36 geographical origins and showed a very high accuracy of 91.86%. The method could be considered as promising for complex food identification, in terms of its reliability, usability, and marker compound independency.

## 2. Results

### 2.1. Comparison of the Integrated HS-SPME Approach with the Conventional LLE Technique to Extract the Chinese Liquor

The headspace solid-phase microextraction (HS-SPME) and liquid-liquid extraction (LLE) are two commonly used methods for extracting compounds from Chinese liquors. We investigated both the HS-SPME and the conventional LLE methods and compared their advantages and disadvantages to evaluate their potentials for extracting the main components from Chinese liquor to the greatest degree possible. We used 50/30-μm divinylbenzene/carboxen/polydimethylsiloxane (DVB/CAR-PDMS) for the HS-SPME and redistilled the diethyl ether and n-pentane for the LLE. The three-dimensional analysis plots of the HS-SPME _DVB/CAR-PDMS_ and LLE of Luzhoulaojiao liquor are shown in Figure 1a, as obtained by the comprehensive two-dimensional GC × GC/TOF-MS. For the Luzhoulaojiao liquor extracted by the HS-SPME and LLE, 1469 and 1532 compounds were identified, respectively, as shown in Figure 1b. The HS-SPME demonstrated the same extraction efficiency, in terms of the GC peak area and the number of identified compounds, as shown by the conventional LLE technique.

The conventional LLE technique requires organic solvents during the extraction, redistilled diethyl ether and n-pentane, which were extracted three times in this research. Based on its potential for sample contamination and loss of analytes during the extraction, the LLE showed its drawbacks in precision. Due to its time consumption, labor intensiveness and limited sensitivity, the LLE is not capable of detecting large numbers of samples. In contrast, the HS-SPME is a solvent-free and sensitive method, and can integrate the sampling, extraction, and concentration into a single step, which is suitable for mass sample detection for the authenticity identification. Thus, the SPME demonstrated a greater sensitivity and simplicity than the LLE. In this study, we chose the HS-SPME to treat the samples for the subsequent procedures.

### 2.2. Comparison of Four Commercial Fibers for Detecting Compounds in the Chinese Liquor

The Partial least-squares (PLS) analysis enabled the practical assessment of the Luzhoulaojiao liquor extraction, and its statistical method of data visualization provided a comparison of the four different fibers and their preferences for compound detection. Using the SPME method discussed in the Method section, a three-component PLS model was established with the parameters R^2^X (0.596), R^2^Y (0.999), and Q^2^ (0.975) for the four different fibers to the areas of the 292 compounds. A score-loading plot (Figure 2) was constructed to visualize the relationship between the fibers and peak areas. The X matrix comprised the colorized box, which denoted the four fibers in triplicate. The Y matrix comprised the black triangles, denoting the areas of the 292 GC × GC/TOF-MS compounds. The areas of the peaks corresponding to the different fibers determined the distribution of the experimental samples on the PLS loading plot. As shown in Figure 2, most areas of peaks were clustered to the left, drifted slightly to the top, and, thus, the most effective extraction fiber was considered to be the 75-μm carboxen/polydimethylsiloxane (CAR/PDMS) fiber. In addition, the results of the SPME experiments performed in triplicate showed good reproducibility in the PLS loading plot except 65-μm PDMS/DVB fiber, which may be the experimental error labeled 1 in the Figure 2.

We compared the identified compounds obtained from the four fibers. Figure 2 illustrates that the four fibers distributed in the different regions showed different preferences for the compound detection. Most of these compounds (e.g., acetals, terpenes, furan, and furanic compounds) were detected using the 75-μm CAR/PDMS fiber, whereas only a portion of these compounds was detected by the other three fibers.

### 2.3. Optimization of the Significant HS-SPME Variables by Plackett-Burman (P-B) Design and Selection of Fiber Coating

Fibers with different types of coating materials have different affinities for various substances. Proper fiber selection plays a key role in providing the optimum extraction conditions for the analysis of volatiles in samples. We chose the four commonly used fibers to compare their ability to extract the Chinese liquor based on the number of peaks identified in the Luzhoulaojiao liquor. A P-B design with ten runs was developed to determine the influence of the experimental variables on the number of chromatographic peaks after applying the SPME conditions detailed in Appendix A. The sample alcohol strength, extraction time, incubation time, extraction temperature, and salt addition were selected for the SPME method optimization. There were significant differences between the diverse fibers based on the number of peaks counted. Figure 3a shows that, considering the four fibers tested, an examination of the total number of peaks for each fiber visibly demonstrated that the most efficient fiber was the 75-μm carboxen/polydimethylsiloxane (CAR/PDMS) fiber, which extracted approximately twice as much as the 85-μm polyacrylate (PA) fiber and more than the 50/30-μm DVB/CAR-PDMS and 65-μm polydimethylsiloxane/divinylbenzene (PDMS/DVB) fibers. The 75-μm CAR/PDMS fiber showed the highest extraction (average number of peaks >3000) and proved to be quite capable at extracting volatiles.

The effects of the variables studied in the example screening experiment are shown in Figure 3b (75-μm CAR/PDMS fiber). The significance of the factors was confirmed by ANOVA, with statistical significance analysis (*p* > 0.05) on the responses of the four fibers. No factors markedly influenced the extraction efficiency response (number of peaks) of the fibers. Only alcohol strength had a significant effect on the response of the PA fiber (Appendix A), while for the other two fibers, there were no factors markedly influencing the extraction efficiency (Appendix A).

Based on the above analysis results, the 75-μm CAR/PDMS fiber demonstrated its advantageous ability to extract the Chinese liquor according to the number of peaks identified. There was no marked difference in the number of peaks identified among the ten SPME conditions. Thus, we chose a more moderate condition for the subsequent Chinese liquor analysis. The optimum values for the variables were set as: 1 M salt addition, 31% alcohol strength, 42.5 °C extraction temperature, 37.5 min extraction time, and 32.5 min incubation time.

### 2.4. Application of the HS-SPME_CAR/PDMS_/GC × GC/TOF-MS Approach for Profiling the Components of Diverse Chinese Liquors

Following the optimization studies, the HS-SPME_CAR/PDMS_/GC × GC/TOF-MS methodology was applied as a reliable alternative to the commonly used LLE, for revealing the main components of Chinese liquors. In total, twelve varieties of Chinese liquors were separated by six different flavor types, which included fen flavor, strong flavor, mixed flavor, sauce flavor, feng flavor, and site flavor, and each flavor type contained two geographical origins for the Chinese liquors. These samples were analyzed using the HS-SPME_CAR/PDMS_ /GC × GC/TOF-MS, and the data obtained were aligned through the LECO ChromaTOF software. Each dataset was processed, and the classification models were built using the partial least-squares (PLS) analysis by soft independent modelling of class analogy (SIMCA), yielding the classification results shown in Figure 4a. Each point on the score scatter plot represents an individual sample, and each sample analysis was repeated three times. The plot with two score components (R^2^X = 0.217, R^2^Y = 0.995, and Q^2^ = 0.924) indicated differences between the diverse flavor types of Chinese liquor. A clear separation was observed between the six flavor types of Chinese liquor. The result revealed that different flavor types of Chinese liquors could be separated according to the different aroma compounds detected by the HS-SPME_CAR/PDMS_/GC × GC/TOF-MS combined with the PLS method. Furthermore, the results for the sample triplicates showed good reproducibility of the PLS loading plot. To obtain the high contribution-degree aromatic compounds, the PLS-VIP (variable importance for projection) was applied to identify and profile the diverse Chinese liquors. To explore the components in the different liquor flavors, we chose compounds with VIP > 1 for processing the following procedures.

Next, the obtained the dataset of VIP values (338 compounds), which were carefully generated through the PLS method described above, was submitted to the hierarchical cluster analysis (HCA) statistical tools to obtain the differential distribution between the six flavor types of Chinese liquors. The numerical data (the standardized peak area of each compound in each sample) was then displayed as a cluster heat map that revealed the hierarchical cluster structure within the data matrix through the rows and columns (Figure 4b). The cluster heat map is composed of differently colored rectangular tiles, the colors of which represent the values of the corresponding elements of the data matrix. In this color scale, red represents a higher peak area, white represents a medium peak area, and blue represents a low peak area. In Figure 4b, two distinctive half-regions that show the differences amongst the compounds of the analyzed samples can be identified based on the row. The upper half-region shows the compounds present in the higher quantities in the site flavor, strong flavor, and sauce flavor groups as compared to the feng flavor, fen flavor, and mixed flavor, but in the lower half-region, the opposite is the case. The columns display different groups that represent the six flavors. The feng, fen, site, and sauce flavors could be clearly identified separately from the mixed and strong flavors, using the HCA clustering. However, the site and strong flavors were slightly confused by the HCA.

### 2.5. Varietal Classification, SVM

For obtaining a classification model for the different geographical origins and flavor types of Chinese liquors, an MS-based database used for the discrimination of samples demonstrated a perfect identification of the geographical origins and flavor types, based on previous studies [24]. Figure 5 shows a total mass spectrum of 262 kinds of Chinese liquors (Appendix A), as obtained by the HS-SPME_CAR/PDMS_/GC × GC/TOF-MS. The total mass spectrum illustrates the complexity and similarity of the samples analyzed, in which 36 geographical origins of Chinese liquors are difficult to distinguish using traditional classification methods, such as the PCA or the PLS. The development of machine learning classification models using the regions (i) *m/z* 20–400 and (ii) *m/z* 20–200 were tested.

The SVM analysis established the composition differences between the different geographical origins and flavor types of Chinese liquors, according to the MS-based database. Thereafter, we used these compositional features for developing MS-based models of the geographical origins and flavor types of Chinese liquors. The results are shown in Appendix A and Figure 6. The best result obtained with the SVM used the mass spectra *m/z* 20–200. The model for the geographical origins and flavor types of Chinese liquors were created using the complete optimization of different parameters/kernels using 176 training datasets. Following the grid search optimization, the optimal cost and gamma values for the model of the geographical origins of Chinese liquors were 16.000 and 0.088, respectively. Using the optimized model to predict 86 test datasets, the prediction accuracy reached 91.86%, using the 10-fold cross-validation technique. This result means that, during the learning process, the optimal distribution of the hyperplane was obtained with the maximum separation distance.

To further investigate the prediction model for the Chinese liquor flavor types, the training and test datasets were classified by their different flavors, including strong flavor, fen flavor, sauce flavor, mixed flavor, site flavor, and feng flavor. Then, after tuning the SVM model settings, the following procedures were performed using the optimized model, and the optimal cost and gamma values were 16.000 and 0.125, respectively. The best results demonstrated that the trained model accuracy was 97.67% with only two test data misclassified, one instance of the sauce flavor and one instance of the mixed flavor were both mistaken for the strong flavor.

To investigate the classification of 86 test datasets of different Chinese liquors by geographical origins or flavor types, we employed a random forest (RF) analysis using *m/z* 20–200, which is an ensemble classifier based on a machine-learning algorithm. The overall out-of-bag (OOB) estimate was 43.18% and 11.36% for the geographical origins and flavors datasets, respectively, suggesting that the similar flavor types from different geographical origins of Chinese liquors were more easily confused. By using the random forest (RF) analysis, the accuracies of the geographical origin and flavor identification were 83.72% and 95.36%, respectively, for the test datasets.

Generally, comparing the two types of machine learning algorithms, the SVM and the random forest (RF) each have their own advantages for classification. In this study, the SVM seems to demonstrate a slight supremacy in both geographical origin and flavor classification. This phenomenon might be caused by the small sample population because the random forest (RF) algorithm should have a large amount of data. However, it also shows that they complement and reinforce each other with the SVM. Interestingly, the random forest (RF) could accurately classify Shuanggou liquor (SG) and Jiannanchun liquor (JNC), that the SVM had missed. Neither the SVM nor the RF could classify Bandaojing liquor (BDJ), Jiangxiaobai liquor (JXB), and Maotaichun liquor (MTC) correctly.

## 3. Materials and Methods

### 3.1. Materials

Sodium chloride, redistilled diethyl ether, n-pentane, and anhydrous sodium sulphate were purchased from Sino-pharm Chemical Reagent Co., Ltd. (Shanghai, China). Water was purified from a Milli-Q system.

### 3.2. Chinese Liquors Samples

A commercial Luzhoulaojiao liquor was used to optimize the conditions of the headspace solid-phase microextraction (HS-SPME) for the detection of compounds.

Two hundred and sixty-two kinds of Chinese liquors were tested in the study from a total of thirty-six geographical origins (Appendix A), which are identified using abbreviations. A total of five kinds of Luzhoulaojiao liquors were supplied from Luzhou Laojiao Co., Ltd. (Luzhou, China). Other liquors were purchased in local supermarkets.

### 3.3. Solid-Phase Microextraction (SPME) Fiber and Parameter

Four commercially available SPME fibers with different stationary phase coating materials and film thicknesses were tested and compared in this study in order to determine which fiber was the most suitable for the sample analyses. The fibers tested in this study, 85-μm polyacrylate (PA), 65-μm polydimethylsiloxane/divinylbenzene (PDMS/DVB), 75-μm carboxen/polydimethylsiloxane (CAR/PDMS), 50/30-μm divinylbenzene/carboxen/polydimethylsiloxane (DVB/CAR-PDMS), were purchased from Supelco (Bellefonte, Pennsylvania, USA). The SPME fibers were conditioned at 250 °C for an hour prior to the analyses, according to the instructions of the manufacturer. 

Each liquor sample (10 mL) was diluted with Milli-Q water (10 mL) to an appropriate concentration of ethanol and then 1 M sodium chloride was added. Ten milliliters of each liquor sample was placed in the 20 mL vial capped with a PTFE septum and aluminum cap. The sample was stirred at a constant speed (100 rpm) during the 32.5 min incubation period, and then extracted for 37.5 min at 42.5 °C in a thermostatic bath. Following that, the fiber was followed by desorption of the analytes into a gas chromatograph injector.

### 3.4. Liquid-Liquid Extraction (LLE)

Two grams of sodium chloride was added to each sample (40 mL), and then extracted four times with the mixture of redistilled diethyl ether and n-pentane, in a reparatory funnel. The organic phase layers were combined and washed with the saturated sodium chloride and deionized water, respectively. Following that, anhydrous sodium sulphate was used to dry the extracts for overnight and then all of the extracts were filtered and concentrated into 0.5 mL, according to the procedure described in literature [25].

### 3.5. Screening of Significant Factor of Using Plackett-Burman Design (P-B Design)

Four fibers were applied for the P-B design to determine how to estimate the significance of five factors. The P-B design was developed in order to reduce the number of experimental runs. It helps recognizing significant variables with fewer experiments, while each variable was observed at two levels (high and low) denoted by (+) and (−), respectively. As shown in Appendix A, a 2 ^ (5-2) P-B design was applied for ten runs to estimate the significance of five variables, for which two replicate at the central point to give a numerical value of the experimental error (pure error). The variables chosen for the SPME optimization were the extraction temperature, the extraction time, the incubation time, the salt addition, and alcohol strength. All of the experiments were carried out in duplicate. Design-Expert software version 8.05b was used to generate the P-B design and for the analyses of the data.

### 3.6. Chromatography

The comprehensive two-dimensional gas chromatography/time-of-flight mass spectrometry (GC × GC/TOF-MS) analyses were performed on an Agilent 7890 gas chromatograph equipped with a TOF-MS (Pegasus 4D, LECO Corporation, Joseph, MI, USA) detector and a LECO ChromaTOF program. The chromatographic separation was achieved using two different polar capillary columns, a conventional nonpolar column DB-5MS (29.950 m × 0.25 mm × 0.25 mm) from J&W Scientific (Folsom, CA, USA) and a medium column DB-17HT (1.640 m × 0.1 mm × 0.1 mm) from the same company. The connection between the two GC columns is a typical combination in GC × GC, which could produce ordered and orthometric chromatograms. According to the previous studies, the initial oven temperature was set to 60 °C for 1 min, then the temperature increased by 5 °C/min up to 165 °C and then increased again by 25 °C/min until it reached the final temperature of 280 °C and then held for 14 min. The mass spectrometer was operated at an acquisition rate of 100 spectra per second, ranging from 20 to 400 u. The electron impact ionization energy was 70 eV and the acquisition voltage was 1700 V. The temperature for the ion source was set to 220 °C.

### 3.7. Data Processing

The obtained raw GC × GC/TOF-MS data were processed and aligned by means of the LECO ChromaTOF software (LECO Corporation, Joseph, MI, USA). The 1st D peak width and the 2nd D peak width were 24 and 0.2, respectively. The automated peak detection and spectral deconvolution with a baseline offset at 1.0 and the signal-to-noise at 50 were used during the data pretreatment. The peak identification was performed using ChromaTOF coupled with the international standard databases (MAINLIB, REPLIB and NIST). Following the data preprocessing, the statistical comparison function in the ChromaTOF software was used to align and compare the data. The resulting three-dimensional dataset comprised the sample information, peak information, and peak intensity. Some artificial peaks generated by noise and column bleed were removed manually from the dataset.

All obtained raw TOF mass spectra were exported from the LECO ChromaTOF software. The baseline subtraction and smoothing were applied for all of the mass spectra. We extracted all of the signals in the range of *m/z* 20–400 with an absolute ion intensity as features for the statistical analysis.

### 3.8. Machine Learning Algorithms

The samples were analyzed using partial least-squares (PLS) and hierarchical cluster analysis (HCA) statistical methods, for an exploratory overview of the different liquor components, and two machine learning algorithms to build statistical models in order to discriminate the studied Chinese liquors according to their geographical origins and flavor types.

A supervised partial least-square (PLS) analysis was performed on the compounds dataset to visualize the different flavor types of the liquor based clustering of the samples and to identify compounds for distinguishing the various liquors. A list of differential compounds was identified based on the variables importance for projection (VIP) score of >1 in the PLS. The VIP is a weighted sum of squares of the PLS-weight, with the weights calculated from the amount of Y-variance of each PLS component. This technique is especially suited to deal with a much larger number of predictors than observations. The PLS analysis was performed using the SIMCA-14.1 software. The PLS model parameters were set for following the statistics that R^2^X is the cumulative modeled variation in the X matrix; R^2^Y is the cumulative modeled variation in the Y matrix; and Q^2^Y is the cumulative predicted variation in the Y variable or matrix, according to the sevenfold cross-validation. The range of these parameters in 0–1, where 1 indicates a perfect fit.

A hierarchical cluster analysis (HCA) was performed to classify the samples in order to show the differential compounds composition based on the similarities of the chemical properties. The results of the PLS analysis form the data matrix as the input of the HCA is associated with the heat map. Each column represented a sample and each row contained the values of the compounds area. The analyses were performed using OmicSolution (http://www.omicsolution.com, accessed on 18 March 2018), a web-based platform for metabolomics data analysis.

The support vector machine (SVM) is a binary classification tool that performs a classification by constructing an optimal separating hyperplane (OSH), which is a supervised learning method that analyzes data and recognizes patterns. The model is constructed by using a non-linear kernel function, which transforms the original space into a high dimensional feature space where the classes can be separated linearly. The radial basis function (RBF) has been used in this study. The SVM was carried out by using the MATLAB R2016b (The Mathworks Inc., Natick, MA, USA). The free SVM toolbox (Zhiren Lin, Taiwan, https://www.csie.ntu.edu.tw/~cjlin/libsvm/, accessed on 20 May 2018) is used in MATLAB to develop the classification models.

The random forest (RF) is a classifier consisting of an ensemble of tree-structured classifiers. The RF algorithm takes advantages of two powerful machine learning techniques: bagging and random feature selection. To assess the prediction performance of the random forest algorithm, it performs a type of cross-validation in parallel with the training step by using the so-called out-of-bag (OOB) samples. On average, each tree is grown using about 2/3 of the training data, leaving about 1/3 as OOB. Because OOB data have not been used in the tree construction, one can use them to estimate the prediction performance. The RF algorithm was implemented by using the random forest R package.

The classification models obtained by the SVM and RF algorithm were externally validated using the same training set (176 training datasets) and test set (86 test datasets), that were manually randomly selected.

## 4. Discussion

Highly developed manufacturing technology promotes the trend of customization, which will greatly increase the diversity and complexity of commercial products [26]. The demand for reliable identification methods for complex products is increasingly urgent, especially for food products, the safety of which is of great concern for many people. Chinese liquor is a typical crucial food commodity that has a complicated chemical composition and faces a serious fraud problem due to its relatively high price [27]. However, the majority of the studies on the analysis of Chinese liquors are primarily based on GC × MS and small datasets, which cannot keep pace with the development of Chinese liquors. Herein, a powerful detection method, GC × GC/TOF-MS, and the efficient machine learning algorithms were integrated to create a comprehensive fingerprinting analysis method that is capable of reliably characterizing Chinese liquors. The prediction model identified as many as 36 geographical origins of Chinese liquors, and the model accuracy was a high as 91.86% and 97.67%. These results show that the GC × GC/TOF-MS was successfully coupled with machine learning for developing reliable food inspection technologies.

The GC × GC/TOF-MS was selected for data collection because of its obvious advantages. It has a greater peak capacity as the result of the combination of multi-dimensional column analysis and the modulation process, which produces narrower and taller chromatographic peaks. With the modulator, small fractions from the first-dimension column were focused and injected into the second-dimension column. The signal in the second-dimension column was enhanced, leading to lower detection limits for chemicals in the complex compound matrix. Because of the special and super column separation system, the chemicals in the Chinese liquors were well-separated, and the MS-based datasets contain almost complete and quite valuable composition information for Chinese liquors. We compared the GC × GC/TOF-MS and GC-MS results using the same sample. The result showed that two columns provided a much greater capacity than one column, and also identified thousands of compounds. The signal of the partial volatiles was increased by two orders of magnitude. It is, therefore, reasonable to suggest that the GC × GC/TOF-MS method put forward in this study will be integral to decomposing the complex compound matrix of Chinese liquor. As a powerful analytical tool, the GC × GC/TOF-MS can produce structured chromatograms even for complex samples. Therefore, we can have visualized insight into the chemical composition of a Chinese liquor using the GC × GC/TOF-MS discussed in this research.

Machine learning techniques play an important role in surpassing the conventional solutions for fingerprinting identification. In particular, machine learning techniques can efficiently process the complicated, highly variable, and nonlinear fingerprinting datasets, and, hence, contribute to solving some of the problems with fingerprinting identification. In this study, we used SVM algorithms to establish a prediction model using 262 MS-based datasets, which classified 36 geographical origins of Chinese liquors for the first time. The optimal values of the model parameters could be determined by grid search optimization. The classification accuracy was evaluated using 86 test datasets, and the accuracy of identification of the geographical origins and flavor types of Chinese liquors were 91.86% and 97.67%, which are very high. The performance of the SVM algorithms showed high prediction accuracy, and it was indicated that the more informative dataset of *m/z* 20–200 could discriminate 36 geographical origins of Chinese liquors. In the studies by Cheng et al. [25], a classification model for eight different geographical origins of Chinese liquors was built. They selected 32 ions from 96 ions as variables by using the partial least squares discrimination analysis (PLS-DA) and the stepwise linear discriminant analysis (SLDA) dimensional reduction methods, and then used them for the discrimination of the eight geographical origins of Chinese liquors. In our study, we selected 181 ions as variables and then used them for the identification of 36 geographical origins of Chinese liquors. There is no doubt that the 181 ions included much more information than that of 32 ions, which may contribute to the high identification accuracy for the enlarged sample size. These results demonstrate that machine learning can efficiently process complicated, highly variable, and nonlinear fingerprinting datasets. It is also notable that only 262 samples were used for training our SVM model. It is reasonable to suggest that the performance of the model would be improved significantly with the gradual enrichment of the training dataset.

As for the SPME, the fiber selection is very important because the various fibers differ in their spectrums of adsorption chemicals [28,29]. Herein, we systematically investigated the effects of the different fibers in term of their sorption ability, which was determined by the peak numbers detected by the GC × GC/TOF-MS. Four fibers (PA, PDMS/DVB, CAR/PDMS, and DVB/CAR/PDMS) were tested for their efficiency in volatile compound extraction. Figure 3a is the comparison of the extraction efficiencies of the four fibers. The results show that the 75-μm CAR/PDMS fiber had the highest extraction efficiency relative to the other fibers, and could detect more than 3000 peaks, well beyond the 2178 peaks of the same liquor detected using 50/30-μm DVB/CAR-PDMS fiber in a recent study [16]. The results suggest that the 75-μm CAR/PDMS fiber has the ability to collect sufficient sample information for achieving a sophisticated high-resolution fingerprinting analysis. Therefore, we could draw the conclusion that the 75-μm CAR/PDMS fiber is suitable for obtaining the sample information for the chemical analysis or identification of Chinese liquors. Additionally, we compared the differences between four commercial fibers for the detection of Chinese liquor compounds (Figure 2). The fibers have preferences for different properties of the analytes. The preference in compound detection is due to the characteristics of the fibers. This implies that a combination of four fibers could detect more compounds.

It was reported that some physical factors affected the SPME extraction efficiency, such as extraction time, extraction temperature, incubation temperature, alcohol strength, ionic strength (salt addition), sample volume, and stirring speed [28,29,30,31,32,33]. Therefore, we systematically investigated the effects of the physical factors on the SPME extraction efficiency. Five previously reported primary impact factors were tested to investigate their contribution to extraction efficiency in the SPME process. A Plackett-Burman (P-B) design was used to determine the primary factor significantly influencing the SPME extraction efficiency. The results showed that the five selected factors had no significant effect on the extraction efficiencies of the PDMS/DVB, CAR/PDMS, and DVB/CAR/PDMS fibers. Alcohol strength had a negative effect on the extraction efficiency of the PA fiber. These results show that physical factors can hardly affect the extraction efficiency, which suggests that the method developed in this study is stable and robust under different external conditions. This will bring obvious advantages to fingerprinting analysis. For example, this will make it easy to develop a standard protocol that can be widely used in different areas by different users. Moreover, a robust method will make the analysis results from different users comparable, which allows the building of a database by collecting data from different labs. It means that different Chinese liquors tested by different researchers could be merged to generate a comprehensive fingerprinting database. In this way, a standardized fingerprinting database can be built, updated, and maintained conveniently. In addition, based on this standardized procedure, we could also develop a fine identification method of Chinese liquors based on a dynamic fingerprinting database.

## 5. Conclusions

In this study, a marker-identification method based on a comprehensive chromatographic analysis and machine learning was developed. We demonstrated that GC × GC/TOF-MS coupled with SPME is a more comprehensive chromatographic method with a greater sensitivity and simplicity than the LLE, which is used for collecting data about the chemical composition of Chinese liquors. Two machine learning algorithms were compared, and the SVM was selected as the preferred identification algorithm, demonstrating supremacy in both geographical origin and flavor classification. We used the developed method for the identification of Chinese liquors from 36 geographical origins, which achieved accuracies of 91.86%. The developed method is considered to be a promising way for building a standardized fingerprint database of Chinese liquors, which would make the web-based comprehensive analysis of Chinese liquors possible. This would finally allow the reliable and convenient inspection of Chinese liquors. 

## Figures and Tables

**Figure 1 molecules-27-06237-f001:**
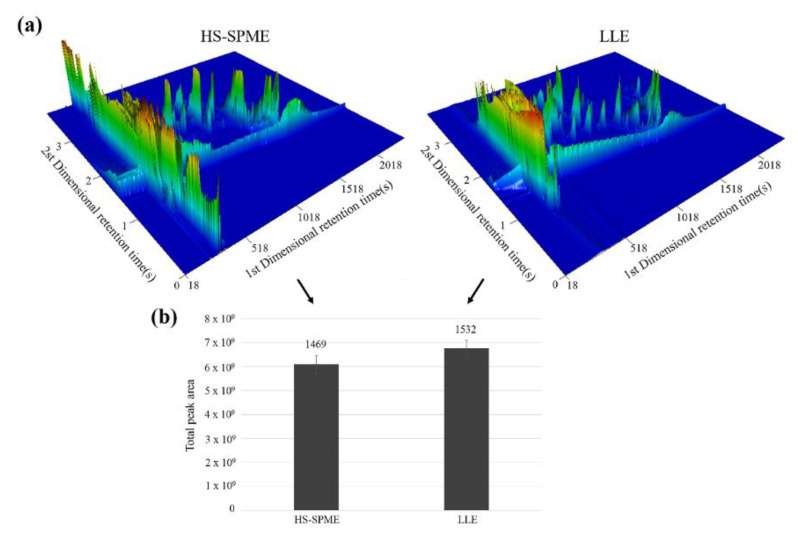
Three-dimensional visualization of raw HS-SPMEDVB/CAR-PDMS /GC × GC/TOF-MS data. (**a**) chromatography 3D plot of Fen liquor extracted by the HS-SPME and (**b**) the LLE. More peaks were detected by the HS-SPMEDVB/CAR-PDMS compared to the LLE, especially, in 17 min in 1st dimensional time according to the 3D plot. Some low abundance peaks invisible using the HS-SPMEDVB/CAR-PDMS could be detected by performing the LLE around 25 min in 1st dimensional time.

**Figure 2 molecules-27-06237-f002:**
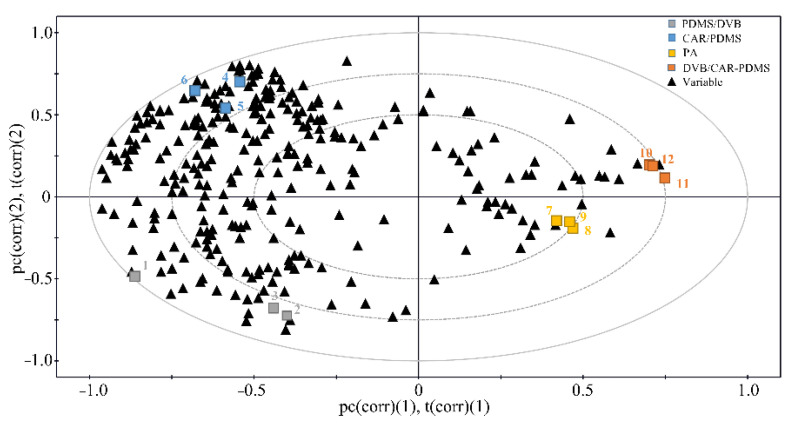
Score-loading biplot of GC × GC/TOF-MS data from the PLS model of Chinese liquor showed the correlation among fibers and the area of 292 compounds. Typical data points representing the experimental solid-phase microextraction fiber. Colorized box representing different fibers and black triangles representing the area of 292 compounds. Each SPME experiment was triplicated.

**Figure 3 molecules-27-06237-f003:**
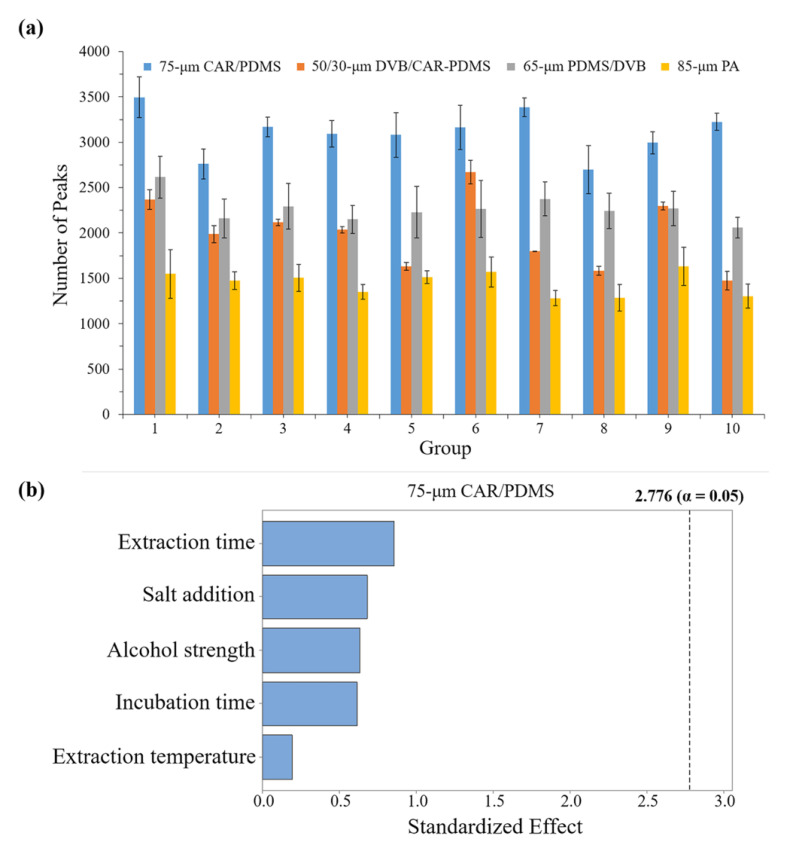
Results of the number of peaks of the different fibers and the Pareto chart of the 75-μm CAR/PDMS. Comparison of the extraction efficiency for each fiber by testing the different groups used in the experiment and measured as number of peaks (**a**). Standardized main effect Pareto chart, representing the estimated effects of the parameters obtained from the P-B design for the determination of the 75-μm CAR/PDMS, Vertical line in the chart defines 95% confidence level (**b**).

**Figure 4 molecules-27-06237-f004:**
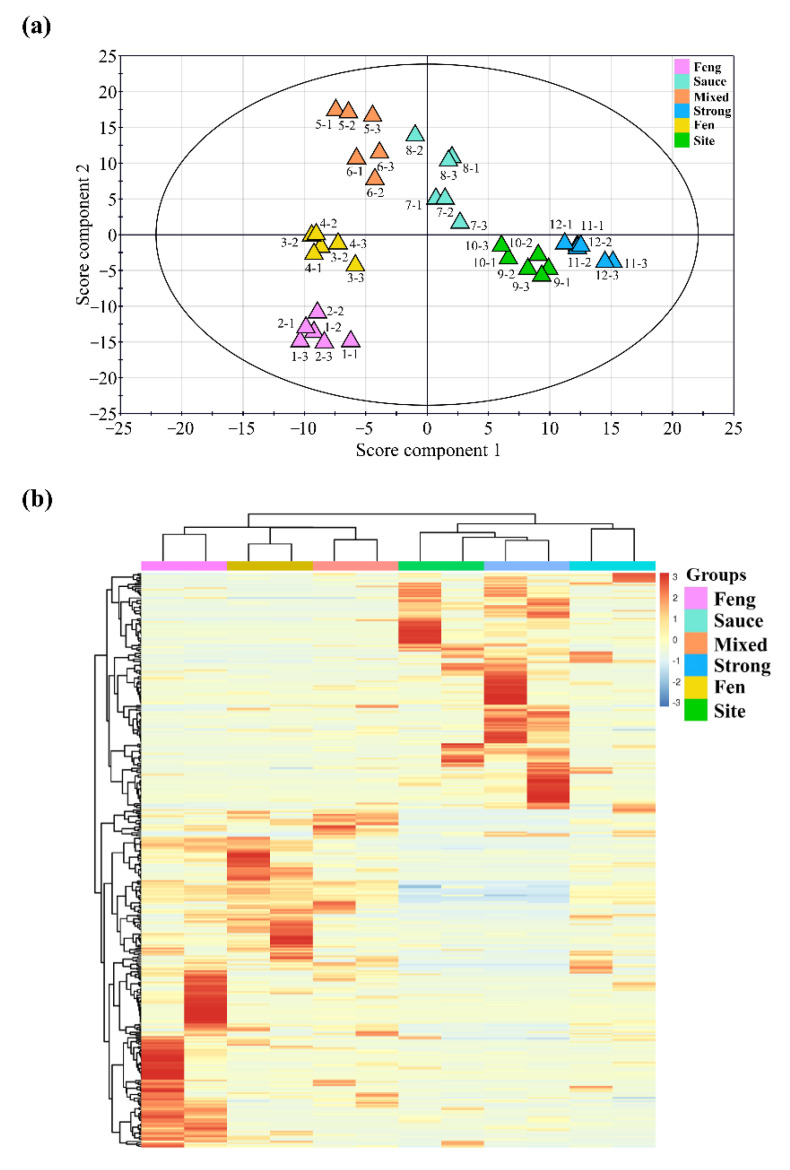
Multivariate statistical analysis of different flavoring types of Chinese liquors. (**a**) PLS score scatter plot of the different flavor liquors based on a correlation analysis. Ellipses and shapes show clustering of the samples. The plot shows a strong correlation between the various flavor liquors. (**b**) Hierarchical clustering heat map of twelve Chinese liquors, with the degree of 339 compounds change in the six different flavoring type liquors. Individual samples (horizontal axis) and compounds (vertical axis) are separated and the top dendrogram is scaled to represent the distance between each branch. Different bar colors represent six types of flavor liquors.

**Figure 5 molecules-27-06237-f005:**
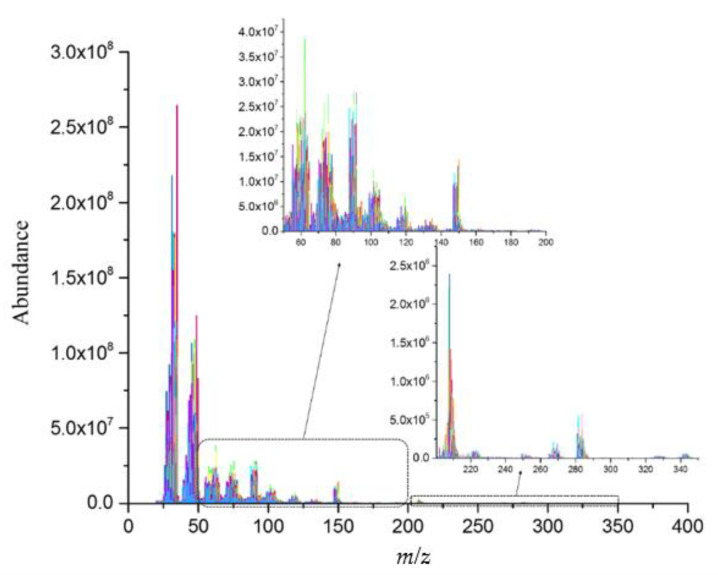
Total mass spectrum of 262 kinds of Chinese liquors obtained using the GC × GC/TOF-MS method.

**Figure 6 molecules-27-06237-f006:**
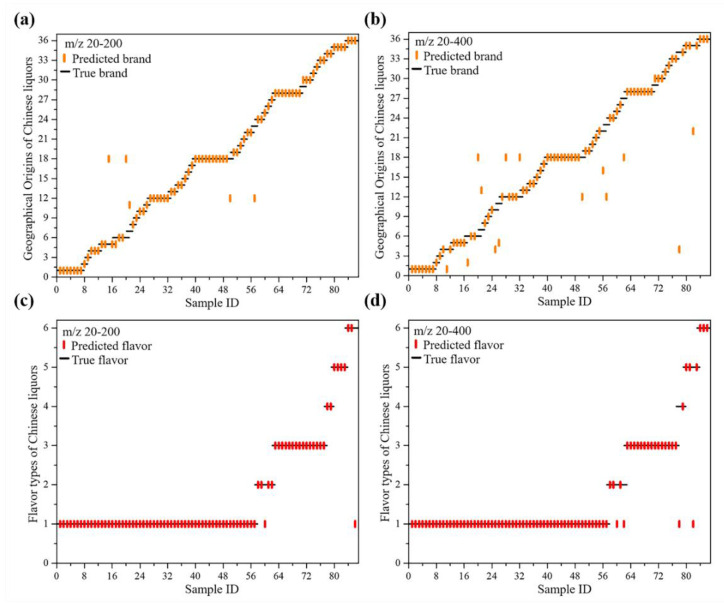
The output results of the SVM machine learning about thirty-six geographical origins of Chinese liquors (**a**,**b**), and six flavor types of Chinese liquors (**c**,**d**).

## Data Availability

Data available from the corresponding authors upon reasonable request.

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
