# Peer review of "Marker-Independent Food Identification Enabled by Combing Machine Learning Algorithms with Comprehensive GC × GC/TOF-MS"

_molecules, 2022, doi:10.3390/molecules27196237_

Round 1

Reviewer 1 Report

The reviewed work has been edited specifically for scientific papers. A fairly comprehensive introduction allows you to get acquainted with its scope. The authors discussed in detail the obtained research results and presented appropriate conclusions. The increasing use of computational methods in analytical chemistry allows for faster obtaining information on the analyzed material, in this case alcoholic beverages. High compliance of the experimental results with those calculated confirms the proper selection of compounds used as determinants of the quality of the liqueurs in question. I find the work interesting and deserves a publication in Molecules.

Reviewer 2 Report

The article covers an interesting topic and proposes a reliable alternative analytical method to identify and evaluate the quality of liqueurs sold and / or produced in China. The results of the article can be used to assess the quality of other alcoholic beverages produced and sold in various places around the world. The method is based on analyzing liqueur samples using the GC × GC / TOF-MS technique and then using two machine learning algorithms to develop and interpret the obtained results. In my opinion, it is the use of machine learning algorithms in the development of results that determines the scientific value of the article and its practical usefulness. The authors have studied two hundred and sixty two kinds of Chinese liquors in the study, in total thirty-six geographical origins of liquors. The article contains a large load of scientific news. The studies are well designed and executed. The research methodology does not raise any doubts. A high-quality chromatographic technique was used and 5 types of SPME fibers were compared. Two separation techniques (SPME and LLE) of analytes from the samples of the tested liqueurs were also compared. The authors collected a large amount of experimental material and subjected it to thorough statistical analysis.

I have two important comments to the article. The first is keywords. In my opinion, they do not fully reflect the content of the work. I suggest replacing them with the following: Chinese liquors; quality; GC × GC / TOF-MS; machine learning;

The second is the conclusions. The first part of the conclusions (lines 520-524) is, in a way, a summary of the article. I believe that it would be better if the conclusions were directly derived from the results obtained. Last sentence of conclusions (lines 529-530) in my opinion, it is too far-reaching and does not result directly from the research carried out.
